TECHNICAL RELEASE

# NeuroVar: an open-source tool for the visualization of gene expression and variation data for biomarkers of neurological diseases

Hiba Ben Aribi[1,*], Najla Abassi[2] and Olaitan I. Awe[3,4]

1 Faculty of Sciences of Tunis, University of Tunis El Manar, 2092, Tunis, Tunisia
2 Laboratory of Biomedical Genomics and Oncogenetics, Institut Pasteur de Tunis, University of Tunis El Manar, 1002, Tunis, Tunisia
3 Department of Computer Science, Faculty of Science, University of Ibadan, 200132, Ibadan, Oyo State, Nigeria
4 African Society for Bioinformatics and Computational Biology, Cape Town, South Africa

## ABSTRACT

The expanding availability of large-scale genomic data and the growing interest in uncovering gene-disease associations call for efficient tools to visualize and evaluate gene expression and genetic variation data. Here, we developed a comprehensive pipeline that was implemented as an interactive Shiny application and a standalone desktop application. NeuroVar is a tool for visualizing genetic variation (single nucleotide polymorphisms and insertions/deletions) and gene expression profiles of biomarkers of neurological diseases. Data collection involved filtering biomarkers related to multiple neurological diseases from the ClinGen database. NeuroVar provides a user-friendly graphical user interface to visualize genomic data and is freely accessible on the project's GitHub repository (https://github.com/omicscodeathon/neurovar).

**Submitted:** 17 August 2024

* Corresponding author. E-mail: benaribi.hiba@gmail.com

Preprint submitted at https://doi.org/10.1101/2024.08.21.609056

Included in the series: *African Society for Bioinformatics and Computational Biology (ASBCB) Omicscodeathon* (https://doi.org/10.46471/GIGABYTE_SERIES_0007)

**Subjects** Software and Workflows, Bioinformatics, Genetics

## STATEMENT OF NEED

Disease biomarkers are genes or molecules that indicate the presence or severity of a disease. Their identification provides important insights into disease etiology and can facilitate the development of new treatments and therapies [1]. Integrating multi-omics data, such as gene expression and genetic variations, has emerged as a powerful approach for biomarker discovery.

Several genomics studies have discovered multiple genetic variations linked to numerous neurological conditions that are complex diseases with a significant level of heterogeneity, such as Alzheimer's disease [2] and Parkinson's disease [3]. Some studies have also used genetic variants to detect the presence of human disorders [4].

The discovered biomarkers are extensively documented in various scientific publications and are accessible through databases like the Clinical Genome (ClinGen) database. ClinGen stores a vast amount of genomic data, including a comprehensive dataset of biomarkers associated with multiple diseases, such as various neurological disorders [5].

Multiple computational tools have been developed in recent years to analyze genomic data, including gene expression data analysis [6, 7], identification of potential inhibitors for

therapeutic targets [8], and comparative analysis of molecular and genetic evolution [9]. However, there is still a need for a specialized tool that focuses on filtering critical disease biomarkers, as this will help in studies that work on finding genes that are involved in diseases using transcriptomic data generated from sequencing experiments [10–13]. Such a tool would help users identify phenotypic subtypes of diseases in their patients, thereby facilitating more accurate diagnoses and personalized treatment plans.

In this study, we developed a novel tool named "NeuroVar" to analyze biomarker data for neurological diseases specifically, including gene expression profiles and genetic variations such as single nucleotide polymorphisms (SNPs) and nucleotide insertion and/or deletion (Indels).

## IMPLEMENTATION

### Data collection

The ClinGen database [5] provides a dataset of biomarkers of multiple diseases from which we filtered data of all the available neurological syndromes (eleven) and non-neurological diseases with neurological manifestations (seven).

### Software development

Two versions of the tool were developed: an R shiny and a desktop application.

The shiny application was developed using multiple R packages, including Shiny (RRID:SCR_001626) [14] and shinydashboard [15]. Other R packages are used for data manipulation, including dplyr (RRID:SCR_016708) [16], readr [17], tidyverse (RRID:SCR_019186) [18], purrr (RRID:SCR_021267) [19], vcfR (RRID:SCR_023453) [20], bslib [21], stringr (RRID:SCR_022813) [22], data.table [23], fs [24], DT [25], sqldf [26], and ggplot2 (RRID:SCR_014601) [27].

For the stand-alone desktop application, the wxPython framework [28] was used to build a similar GUI. A variety of Python libraries were employed, including Pandas (RRID:SCR_018214) [29], MatPlotLib (RRID:SCR_008624) [30], and NumPy (RRID:SCR_008633) [31]. After testing, the application was packaged as an installer using cx_Freeze [32]. Finally, it was distributed as a zip file to be downloaded.

### Pipeline validation and case study

To validate the pipeline, a case study was performed on the public dataset SRP149638 [33] available on the SRA database [34]. The dataset corresponds to RNA sequencing data from the peripheral blood mononuclear cells from healthy donors and Amyotrophic Lateral Sclerosis (ALS) patients. The ALS patients involved in the study have mutations in the FUS, TARDBP, SOD1, and VCP genes.

The file's preprocessing, genetic expression analysis, and variant calling were performed using the Exvar R package [35]. The Exvar package uses the rfastp package [36] and the gmapR package [37] for preprocessing fastq files, the GenomicAlignments package (RRID:SCR_024236) [38], and the DESeq2 packages (RRID:SCR_015687) [39] for gene expression data analysis, as well as the VariantTools [40] and VariantAnnotation (RRID:SCR_000074) [41] packages for variant calling.

## RESULTS

### Supported disease

NeuroVar integrates biomarkers of multiple neurological diseases, including epilepsy, ALS, intellectual disability, autism spectrum disorder, brain malformation syndrome, syndromic disorders, cerebral palsy, RASopathy, aminoacidopathy, craniofacial malformations, Parkinson's disease, and PHARC syndrome. It also integrates seven non-neurological diseases with neurological manifestations: peroxisomal disorders, hereditary cancer, mitochondrial disease, retina-related disorders, general gene curation, hearing loss, and fatty acid oxidation disorders. Each disease syndrome includes multiple disease types; for example, sixteen types of ALS disorder are integrated.

### Operation and implementation

The desktop and Shiny applications have the same user interface; however, the implementation is different.

The Shiny application is platform-independent, while the desktop application is optimized for the Windows operating system. The necessary library requirements for the tool are automatically installed in both versions. The amount of RAM used depends on the servers or the machine being used, and the only prerequisites for using the tool are having R installed for the shiny application and having Python installed for the desktop application.

The tool is compatible with RNA sequencing data. The input data files should be in CSV format for gene expression data and VCF (Variant Call Format) format for genetic variants data. Guidance of the files' organization is available in the tool's Github repository in detail (path: omicscodeathon/neurovar/demonstration_data).

Detailed guidelines for installing and using both versions of the application are provided in the project's GitHub repository.

### The application's usage

The application dashboard includes three pages. The first page, named "Biomarker", provides data on the disease's biomarkers. Initially, the user should select the target disease syndrome and the specific disease subtypes from the provided list (Figure 1).

Next, a list of biomarkers is provided with additional data, including the gene's mode of inheritance, description, type, and transcripts. Also, a link for the official online report validating the gene's association with the disease is provided (Figure 2).

The second page, named "Expression", is used to visualize the biomarkers expression profile. After importing a CSV file and identifying the key columns, the log2FC value and adjusted *p*-value are requested to define the differential expression profile. By default, the adjusted *p*-value is set to less than 0.01, and the logFC value is set to less or more than 2 (Figure 3).

As a result, the expression profiles of the target disease biomarkers (previously selected) are summarized in a table and represented in a volcano plot (Figure 4).

The third page, named "Variants", allows the visualization of SNPs and Indels data. The user is requested to define the path to the directory containing the VCF files. The files are expected to be divided into two folders, named "controls" and "patients", containing the VCF files of the controls and patients, respectively. The user needs to define the variant type as SNPs or Indels (Figure 5).



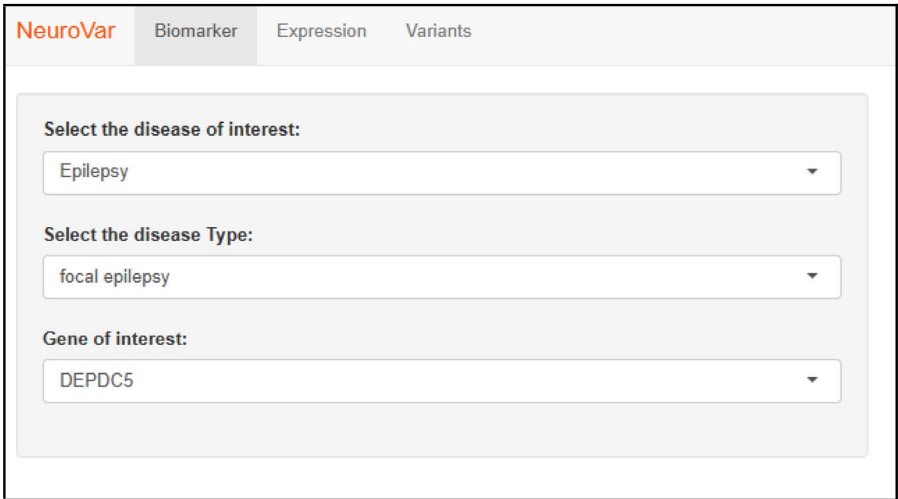

**Figure 1.** The layout of the "Biomarker" page. The user is requested to define the target disease, disease type, and gene of interest.

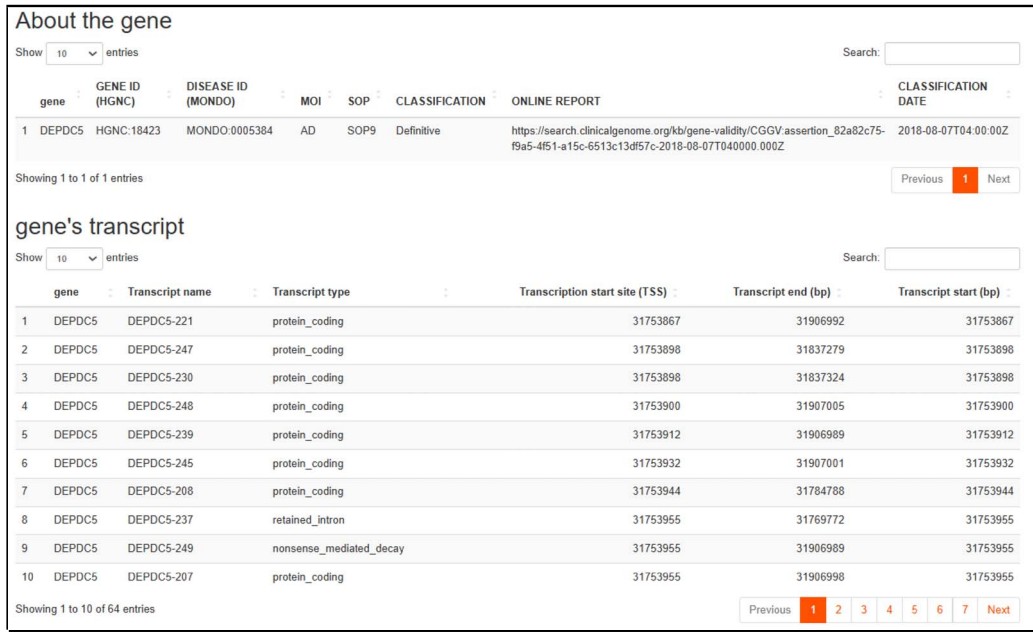

**Figure 2.** The output of the "Biomarker" page. The output includes two tables detailing key information about the selected gene.

The VCF files are processed and annotated, and then the variants in the target disease biomarkers are filtered and resumed in a table comparing the reference genome, the control group, and the patients' group (Figures 6 and 7).



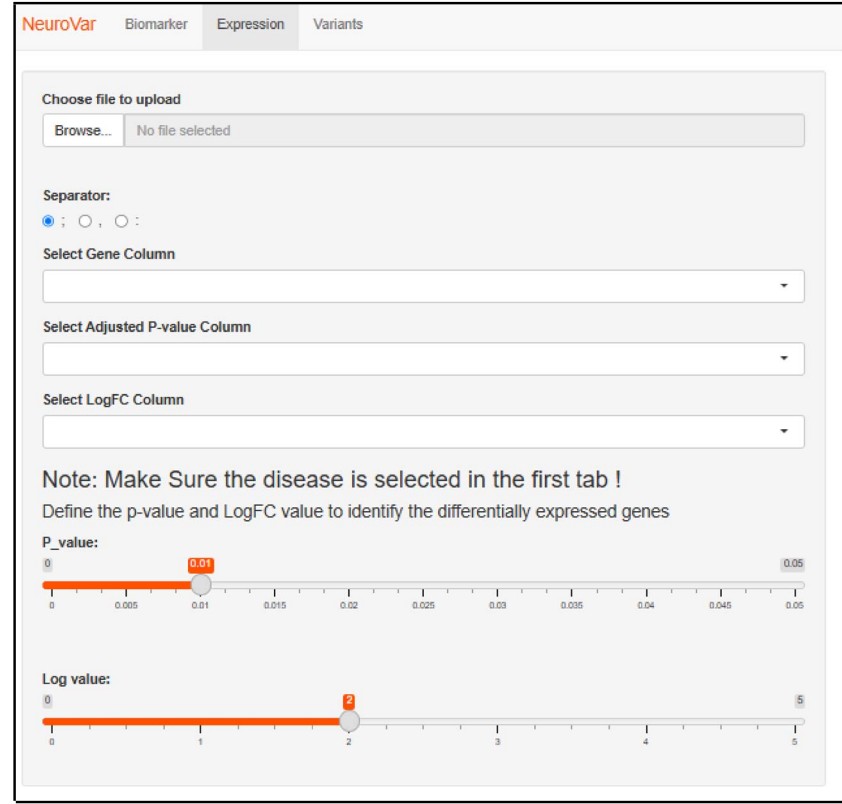

**Figure 3.** The layout of the "Expression" page. The user is requested to upload the data file and select the *p*-value and the log-FC value required to construct the differential expression profile.

## Case study results

To validate the pipeline, we conducted a case study using the public dataset that provides RNA sequencing data of ALS patients who were declared to carry mutations in the FUS, TARDBP, SOD1, and VCP genes [33].

Initially, we used NeuroVar to explore the roles of the genes FUS, TARDBP, SOD1, and VCP in ALS. Our findings confirmed that FUS, TARDBP, and SOD1 are recognized ALS biomarkers, while VCP is not. ALS has 26 subtypes, with FUS being a biomarker for type 6, SOD1 for type 1, and TARDBP for type 10, suggesting that the patients in the study may represent a mixture of these ALS subtypes.

Next, we investigated whether mutations in these genes impacted their expression profiles. Using an adjusted *p*-value threshold of 0.05 and a log fold change (logFC) cutoff of 2, we found that out of 21 known ALS biomarkers, only one gene—TUBA4A—was differentially expressed. Notably, none of the four genes (FUS, TARDBP, SOD1, and VCP) showed differential expression.

Finally, we examined the types of mutations present in these genes. We detected 23 SNPs across seven biomarkers: DAO (all ALS types), FIG4 (ALS type 11), ERBB4 (ALS type 19), TUBA4A (ALS type 22), KIF5A (ALS type 25), C9orf72 (ALS type 1), and TBK1 (ALS type 4). No indels were detected in any of the biomarkers. Interestingly, the biomarkers FUS, TARDBP,

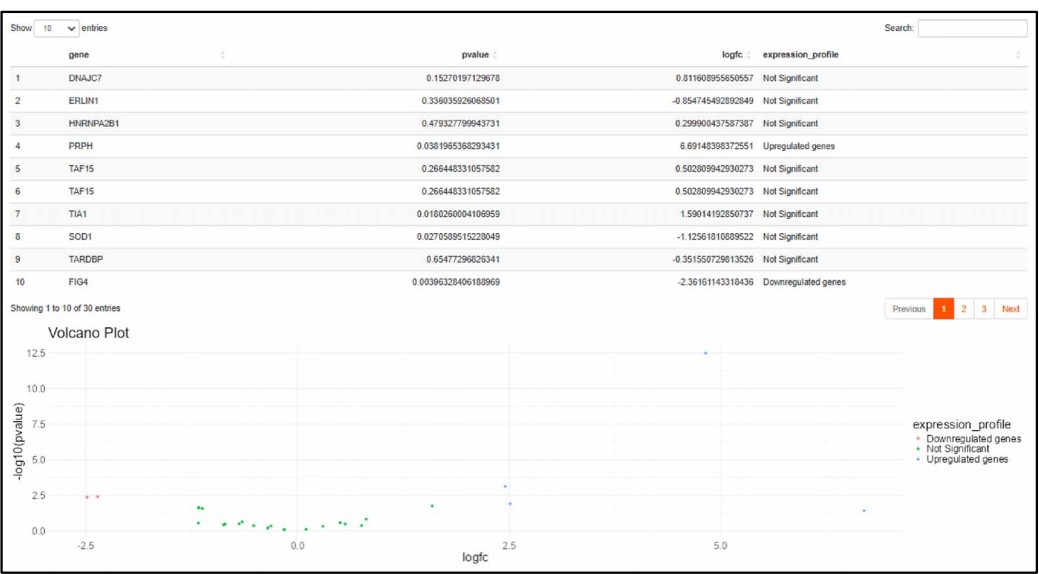

**Figure 4.** The output of the "Expression" page. As output, a summary of the genes' expression profiles is displayed in a table and a volcano plot.

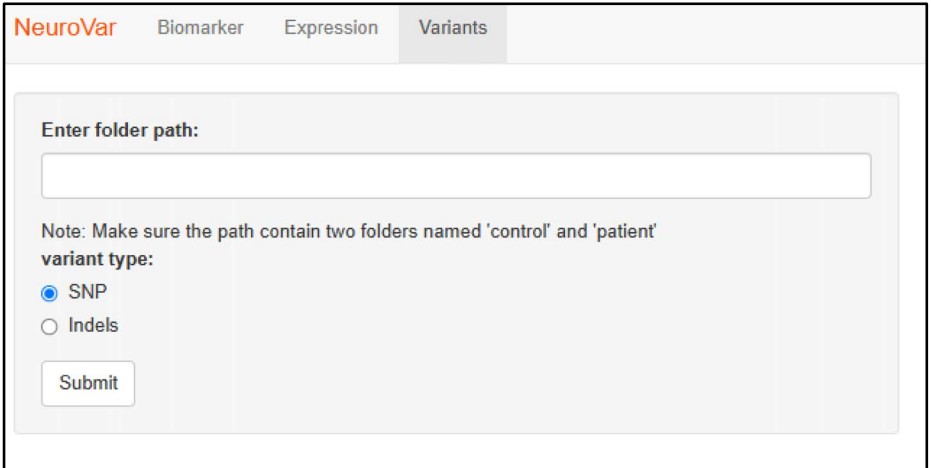

**Figure 5.** The layout of the "Variant" page. The user is prompted to specify the path to the data-containing folder and the data type.

and SOD1 exhibited neither SNPs nor Indels, suggesting that the mutations in these genes may be due to other types of genomic changes.

A demonstration video describing how to visualize the demonstration data using neurovar is available on GitHub (Figure 8).

## DISCUSSION AND CONCLUSION

NeuroVar is a novel tool for visualizing genetic variation and gene expression data related to neurological diseases. The tool is designed to visualize genetic variation and gene

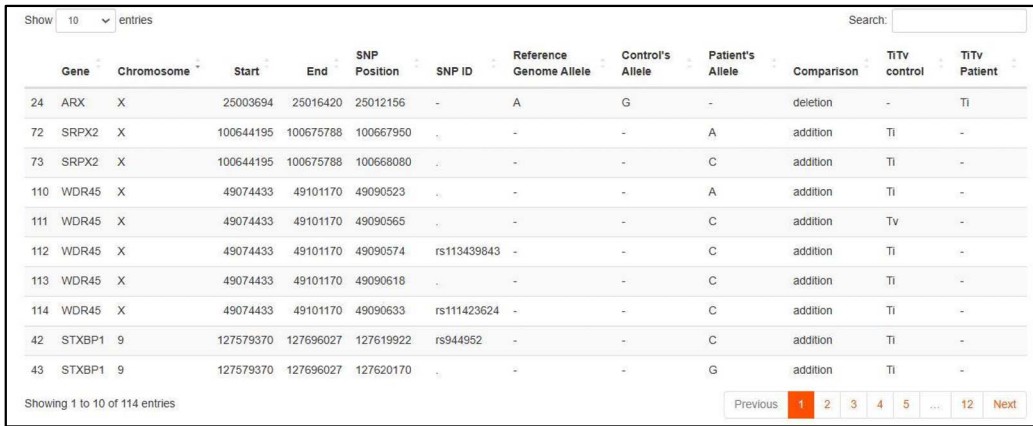

**Figure 6.** The output of the "Variant" page. Table summarizing the SNPs in the target disease's biomarkers.

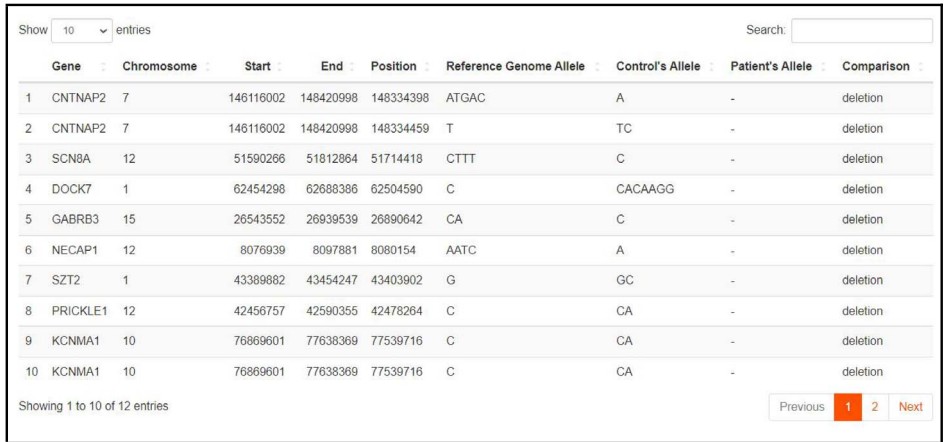

**Figure 7.** The output of the "Variant" page. Table summarizing the INDELs in the target disease's biomarkers.

expression data, with a particular emphasis on neurological disorders. This specialization makes it an invaluable resource for researchers and clinicians focused on these conditions. It offers features to filter biomarkers by specific diseases, which aids in confirming gene-disease associations and prioritizing genes for further investigation.

The tool supports eleven neurological syndromes and seven non-neurological diseases with neurological manifestations. While the supported diseases list is currently limited to data from the ClinGen database, it will be frequently updated, and data sources will be expanded to include other databases in the future.

NeuroVar is available as a desktop application and as a Shiny application. Both versions are user-friendly and do not require computational skills to operate them. Additionally, all necessary dependencies are automatically installed with the tools. This dual accessibility of NeuroVar caters to users with varying preferences and technical backgrounds, which makes it more accessible and easier to use than other visualization tools of genetic variant data, such as the command line tool VIVA [42] to analyze VCF files and the "Transcriptomics



**Figure 8.** Video demonstration of the NeuroVar Shiny Application [44]. https://youtu.be/cYZ8WOvabJs?si=W7v3AZ_pAsXt7ZsI.

oSPARC" web tool for gene expression data visualization hosted on the o$^2$S$^2$PARC platform (RRID:SCR_018997) [6].

In addition to its user-friendly design, NeuroVar streamlines the research workflow by eliminating the need for multiple filtering steps across different platforms. By integrating essential functions within a single interface, it allows users to conduct comprehensive analyses without leaving the application, thereby enhancing efficiency and reducing errors. The inclusion of a quick-access library on the first page further aids in referencing important data, making it easier to revisit and validate findings. This centralization of tasks, coupled with a focus on neurological diseases and extensive biomarker information, makes NeuroVar a highly useful tool for advancing research in the field.

## AVAILABILITY OF SOURCE CODE AND REQUIREMENTS

- Project name: NeuroVar
- Project home page: https://github.com/omicscodeathon/neurovar
- Operating system: Platform independent
- Programming language: Python and R
- Other requirements: None

- License: Artistic license 2.0
- RRID: SCR_025640
- DOI for the Project's GitHub Repository: https://doi.org/10.5281/zenodo.13375646
- DOI for the Shiny application: https://doi.org/10.5281/zenodo.13375493
- DOI for the desktop application: https://doi.org/10.5281/zenodo.13375579
- DOI for the data: https://doi.org/10.5281/zenodo.13375591.

## DATA AVAILABILITY

The following resources can be accessed in the project's GitHub repository, https://github.com/omicscodeathon/neurovar:

- The open-source code for both the Shiny application and the desktop application.
- An installation guide.
- A video demonstration.
- The processed case study data is available as demonstration data in Zenodo [43].

Data came from ClinVar, and the presented case study was performed on the public dataset SRP149638 from the SRA database.

The open source code of the Shiny application and the desktop application are available in the project's GitHub Repository: https://github.com/omicscodeathon/neurovar.

Installation Guide, demonstration data, and video demonstration (Figure 8) are also available in the project's GitHub Repository: https://github.com/omicscodeathon/neurovar.

Snapshots of the project code [45], shiny application code [46], and desktop application code [47] are all in Zenodo.

## ABBREVIATIONS

ALS: Amyotrophic Lateral Sclerosis; Indel: insertion and/or deletion; logFC: log fold change; SNP: single nucleotide polymorphism; VCF: Variant Call Format.

## DECLARATIONS

### Ethics approval and consent to participate

The authors declare that ethical approval was not required for this type of research.

### Consent for publication

Not applicable.

### Competing interests

The authors declare that they have no competing interests.

### Authors' contributions

HBA: Conceptualization, Methodology, Validation, Writing; NA: Data Analysis, Methodology, Validation, Writing; OIA: Resources, Manuscript review, and Project Supervision.

### Funding

The authors declare that no financial support was received for the research, authorship, and/or publication of this article.

## Acknowledgements

The authors thank the National Institutes of Health (NIH) Office of Data Science Strategy (ODSS), and the National Center for Biotechnology Information (NCBI) for their immense support before and during the April 2023 Omics codeathon organized by the African Society for Bioinformatics and Computational Biology (ASBCB).

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
