## [Editor Report]

Editor’s AssessmentCoded and written up as part of the African Society for Bioinformatics and Computational Biology (ASBCB) Omicscodeathons, NeuroVar is a new tool for visualizing genetic variation (Single nucleotide polymorphisms and insertions/deletions) and gene expression data related to neurological diseases. The open source R-tool is available as an online Shiny Application and a desktop application that does not require any computational skills to use. Initial validation and case studies for the tool present analyses of biomarkers in ALS, exemplifying its relevance in personalized medicine and genomic discovery. Being an Open Source project, after peer review more detail has been added in paper and GitHub repo on how to contribute, report issues or seek support. Alongside some improved installation guidelines. The paper states future developments will expand its dataset beyond the ClinGen database to encompass new databases and broader genetic inquiries.Editor’s AssessmentCoded and written up as part of the African Society for Bioinformatics and Computational Biology (ASBCB) Omicscodeathons, NeuroVar is a new tool for visualizing genetic variation (Single nucleotide polymorphisms and insertions/deletions) and gene expression data related to neurological diseases. The open source R-tool is available as an online Shiny Application and a desktop application that does not require any computational skills to use. Initial validation and case studies for the tool present analyses of biomarkers in ALS, exemplifying its relevance in personalized medicine and genomic discovery. Being an Open Source project, after peer review more detail has been added in paper and GitHub repo on how to contribute, report issues or seek support. Alongside some improved installation guidelines. The paper states future developments will expand its dataset beyond the ClinGen database to encompass new databases and broader genetic inquiries.

---

## [Reviewer Report]

Reviewer name and names of any other individual's who aided in reviewerJoost WagenaarDo you understand and agree to our policy of having open and named reviews, and having your review included with the published manuscript. (If no, please inform the editor that you cannot review this manuscript.)YesIs the language of sufficient quality?YesPlease add additional comments on language quality to clarify if neededIs there a clear statement of need explaining what problems the software is designed to solve and who the target audience is? YesAdditional CommentsThere is a clear statement of need, but the audience is not very targeted. The investigators outline the need for tools to help users identify phenotypic subtypes of disease and describe how the tool would help with this. Although the investigators mention that the tool will allow users to analyze biomarker data, the scope of the types of analysis that can be performed is relatively small. I think that it would benefit the tool to better define the targeted users (clinicians, data scientists, enthusiasts?) and develop specifically towards a single audience. The tool leverages several existing R packages to run the analysis over the data and the provided tool can be described as a user-friendly wrapper around these libraries. The interface allows users to submit a file, and plot the results of the analysis within the app.Is the source code available, and has an appropriate Open Source Initiative license <a href="https://opensource.org/licenses" target="_blank">(https://opensource.org/licenses)</a> been assigned to the code?YesAdditional CommentsAs Open Source Software are there guidelines on how to contribute, report issues or seek support on the code?NoAdditional CommentsI did not see any guidelines for contributing to the project in the paper, or in the associated GitHub repository.Is the code executable?YesAdditional CommentsIs installation/deployment sufficiently outlined in the paper and documentation, and does it proceed as outlined?YesAdditional CommentsIs the documentation provided clear and user friendly?YesAdditional CommentsYes, the investigators did a great job providing documentation and installation instructions.Is there enough clear information in the documentation to install, run and test this tool, including information on where to seek help if required?YesAdditional CommentsIs there a clearly-stated list of dependencies, and is the core functionality of the software documented to a satisfactory level?YesAdditional CommentsYes, the investigators provide a clearly-stated list of dependencies and instructions on how to install them prior to running the application.Have any claims of performance been sufficiently tested and compared to other commonly-used packages? YesAdditional CommentsIs test data available, either included with the submission or openly available via cited third party sources (e.g. accession numbers, data DOIs)?YesAdditional CommentsThe paper, and GitHub repository point to a public dataset that can be used to test the application.Are there (ideally real world) examples demonstrating use of the software? YesAdditional CommentsThe investigators provide a video highlighting the use of the application and provide a use-case where they use the app to validate some existing knowledge.Is automated testing used or are there manual steps described so that the functionality of the software can be verified?NoAdditional CommentsThe application is sufficiently small that no automated testing or manual testing would necessary be required beyond validating that the application works.Any Additional Overall Comments to the AuthorThe proposed application provides a nice tool that makes visualization of vcf data and analysis easier for users who are not comfortable working within R directly. It provides a nice demonstration how the scientific community can wrap scientific tools into deployable applications and tools that can be easily understood. A question remains on the target audience for an application like this as most people who are interested in these type of analysis and visualizations are, in fact, familiar enough with R, or other programming languages to directly leverage the libraries and plot the results. That said, as data integration and multi-omics visualization becomes more complex and the app provides more ways to visualize the data in meaningful ways, I do strongly believe that applications like this can provide a meaningful addition to the scientific tools that are available.RecommendationAccept

---

## [Reviewer Report]

Reviewer name and names of any other individual's who aided in reviewerRuslan RustDo you understand and agree to our policy of having open and named reviews, and having your review included with the published manuscript. (If no, please inform the editor that you cannot review this manuscript.)YesIs the language of sufficient quality?YesPlease add additional comments on language quality to clarify if neededThe language quality of the document is of sufficient quality. I did not notice any major issues.Is there a clear statement of need explaining what problems the software is designed to solve and who the target audience is? YesAdditional CommentsYes, authors provide a statement of need. Authors mention that there is the need for a specialized software tool to identify genes from transcriptomic data and genetic variations such as SNPs, specifically for neurological diseases. Perhaps authors could expand on how they chose the diseases. E.g. stroke is not listed among the neurological diseases. Perhaps authors could expand a bit on the diseases they chose in the introduction.Is the source code available, and has an appropriate Open Source Initiative license <a href="https://opensource.org/licenses" target="_blank">(https://opensource.org/licenses)</a> been assigned to the code?YesAdditional CommentsYes the source code is available in github under the following link: https://github.com/omicscodeathon/neurovar. Additionally authors deposited the source code and additional supplementary data in a permanent depository with zenodo under the following DOI: https://zenodo.org/records/13375493. They also provided test data https://zenodo.org/records/13375591. I was able to download and access the complete set of dataAs Open Source Software are there guidelines on how to contribute, report issues or seek support on the code?NoAdditional CommentsI did not find any way to contribute, report issues or seek support. I would recommend that the authors add this information to the Github README file.Is the code executable?YesAdditional CommentsYes, I could execute the code using Rstudio 4.3.3Is installation/deployment sufficiently outlined in the paper and documentation, and does it proceed as outlined?YesAdditional CommentsI could follow the installation process, but perhaps authors could add few more details how to download from Github in more detail. As some scientist may have trouble with it. Also perhaps an installation video (additionally to the video demonstration of the Neurovar Shiny App might be helpful.·Is the documentation provided clear and user friendly?YesAdditional CommentsThe documentation is provided and is user friendly. I was able to install, test and run the tool using RStudio. Authors may consider to offer also a simple website link for the RshinyTools if possible. This may enable the access also for scientists that are not familiar with R.Especially, it is great that authors provided a demonstration video. I was able to reproduce the steps. However, I would recommend to add more information into the Youtube video. E.g. reference to the preprint/ paper and Github link would be helpful to connect the data.Perhaps authors could also expand a bit on the possibilities to export data from their software. And provide different formats e.g., PDF / PNG /JPEG. I think this is important for many researchs to export their outputs e.g., from the heatmaps.Is there enough clear information in the documentation to install, run and test this tool, including information on where to seek help if required?YesAdditional CommentsIs there a clearly-stated list of dependencies, and is the core functionality of the software documented to a satisfactory level?YesAdditional CommentsYes, dependencies are listed and are installed automatically. It worked for me with Rstudio version 4.3.3. In the manuscript and in the repository.Have any claims of performance been sufficiently tested and compared to other commonly-used packages? Not applicableAdditional CommentsIs test data available, either included with the submission or openly available via cited third party sources (e.g. accession numbers, data DOIs)?YesAdditional CommentsYes the authors provide test data with this doi: https://doi.org/10.5281/zenodo.13375590Are there (ideally real world) examples demonstrating use of the software? YesAdditional CommentsYes, authors use the example of Epilepsy, focal epilepsy and the gene of interest DEPDC5. I replicated their search and got the same results. However, I find that the label in Figure 1 in the gene’s transcript could be a bit more clear. E.g. it is not clear to me what transcript start and end refers to. It might also be more helpful if authors provide an example dataset for the Expression data that is loaded in the software by default.Furthermore authors use a case study results using RNAseq in ALS patients with mutations in FUS, TARDBP, SOD1, VCP genes.Is automated testing used or are there manual steps described so that the functionality of the software can be verified?NoAdditional CommentsAutomated testing is not used as far as I can access it.Any Additional Overall Comments to the AuthorThe preprint version of this paper was also reviewed in ResearchHub: https://www.researchhub.com/paper/7381836/neurovar-an-open-source-tool-for-gene-expression-and-variation-data-visualization-for-biomarkers-of-neurological-diseases/reviews My expertise: I am assistant professor in neuroscience and physiology at University of Southern California and work on stem cell therapies on stroke. We are particularly interested in working with genomic data and the development of new biomarkers for stroke, AD and other neurological diseases. Summary: The authors provide a software tool NeuroVar that helps visualizing genetic variations and gene expression profiles of biomarkers in different neurological diseases.RecommendationMinor Revisions